# Towards Understanding Subsurface Characteristics in Burn Process of Gear Profile Grinding

**DOI:** 10.3390/ma16062493

**Published:** 2023-03-21

**Authors:** Jun Wen, Jinyuan Tang, Wen Shao, Weihua Zhou, Weiwei Huang

**Affiliations:** 1State Key Laboratory for High Performance Complex Manufacturing, Central South University, Changsha 410083, China; 2College of Mechanical and Electrical Engineering, Central South University, Changsha 410083, China; 3Powder Metallurgy Research Institute, Central South University, Changsha 410083, China

**Keywords:** gear grinding, grinding burn, surface integrity, microstructure evolution

## Abstract

In gear profile grinding, the grinding burn will greatly influence the anti-fatigue performance of gears. However, the influence of microstructure change caused by grinding burn on gear surface integrity is still unclear. In this paper, full-factor experiments of gear profile grinding are conducted and the grinding temperature is measured during the experiments. Furthermore, the tooth surface integrity including microstructure, residual stress, microhardness, and surface morphology is characterized. The relationship between grinding parameters, grinding burns and subsurface layer properties is analyzed by systematical test results. Radial grinding depths of more than 20 μm matched with wheel speeds below 30 m/s will result in severe grinding burns. The effect of grinding burns on the grain state mainly results in the breakdown of high strength martensite and the formation of inhomogeneous secondary tempered sorbite. The recovery and recrystallization of the microstructure of the tooth subsurface layer after grinding burns is the root cause of the substantial reduction in compressive residual stress and nano-hardness. The occurrence of grinding burns is mainly due to the unreasonable matching of process parameters rather than being influenced by a single grinding parameter alone. The risk of burn can be significantly reduced at greater wheel speeds and lower radial grinding depth. This study presents an insight into the mechanism of the effect of gear profile grinding burns on the surface integrity of the tooth flank.

## 1. Introduction

Gear is a key component of transmission system, which is widely used in many fields such as aerospace, engineering machinery, precision instruments. As gear systems are used in more extreme and demanding environments, higher demands are placed on the service performance of gears. In addition to consideration of tooth profile, gear design based on topology optimization and machine learning is often used to improve the design strength of gears [1,2]. In recent years, computational fluid dynamics has been commonly used to study issues such as friction, losses, and transmission efficiency in gears, and it can also be used in the flow analysis of gear lubricants to optimize the lubrication performance of gears [3]. Extensive research has also been carried out on gear working wear, with in-depth discussions on gear wear mechanisms, wear evaluation methods, and wear prediction and control [4,5,6,7]. Tooth wear is mainly caused by small plastic deformations caused by contact stress on the tooth surface and crack expansion caused by surface fatigue. The number, shape, and size of wear particles are usually used to evaluate the wear degree of gears, and this method has the characteristics of repeatability and high accuracy. The design and manufacture of gear are a complete system in the gear development process and have an inseparable relationship. Gear profile grinding is the essential procedure in the gear manufacturing chain and has a major influence on the resultant workpiece properties [8,9,10,11,12]. Due to the complex wheel–tooth movement relationship and the uneven distribution of thermal loads on the tooth flank, the probability of grinding burns in gear profile grinding is significantly higher than that of surface grinding. The occurrence of grinding burns can directly affect the surface integrity of the gear including microstructure, residual stresses, and hardness. Hence, it is important to understand the subsurface layer properties alteration in burn process of gear profile grinding.

Many theoretical and experimental studies have also emerged in recent years in the study of gear grinding burns and microstructure evolution. Su et al. [13] developed an ANSYS software-based model of gear profile grinding heat parameters to study the temperature of grinding burns. Subsequently, metallographic experiments were carried out to further investigate the tooth flanks after grinding burns and to propose solutions to prevent grinding burns. Heinzel et al. [14] analyzed the thermal and depth effects in different grinding processes by means of experiments, using specific grinding power and contact time to characterize a process lower limit for the occurrence of grinding burns with different kinematics. The effect of the grinding process on the microstructure was an important part of the work by Hüsemann et al. [15]; the metallographic results after grinding showed no significant difference between the metallographic organization of the tooth surface after grinding and before grinding at lower spindle power, and a large change in the tooth surface metallography after grinding at higher spindle power. Guerrini et al. [16] proposed a dry grinding process for gears that does not depend on grinding fluid. The parameter thresholds for avoiding grinding burns were first determined by metallographic analysis, and then the parameters were optimized to improve the machining accuracy of gears. In order to specifically study the effects caused by grinding burns, Dychtońet et al. [17] used laser heating to simulate the effect of excessive tempering and secondary hardening caused by grinding burns in aerospace gears and investigated the effect of burns on microstructure and residual stresses by Barkhausen noise nondestructive testing. For high-strength gear steel materials, Wang et al. [18] conducted an experimental study on the characteristics and formation mechanisms of grinding burns and cracks in 20 CrMnTi steel gears, and analyzed the changes in tissue transformation, hardness, and residual stresses for tempered burns and quenched burns, respectively. Kang et al. [19] studied the evolution of microstructure and residual stresses in 17 CrNiMo6 steel gears under different grinding parameters. The microstructure before and after grinding was observed by transmission electron microscope (TEM), and it was found that the original boundaries of ferrite and martensite phases became blurred and some dislocations appeared in the structure. Although the research object is not gears, there are many excellent studies on grinding microstructure evolution by Xiu and Sun et al. For example, the effect of grinding chatter on martensite nucleation and growth and the mechanism of grain refinement in the grinding metamorphic layer were investigated by combining the calculation of the grinding temperature field with a hybrid model of metamorphic automata [20,21]. A digital twin model was proposed to study the evolution of grinding temperature and microstructure along the surface depth direction from a microscopic perspective [22]. The effect of pre-stress and grinding parameters on the dynamic recrystallization behavior of grinding in AISI 1045 steel was investigated [23]. A new model for the prediction of dynamic properties, mechanical–thermal interactions, and transformation effects was proposed in terms of microstructure and residual stresses [24].

In addition to research on the hazards of grinding burns, the prevention and detection of grinding burns is also an important research task. Based on the use of Barkhausen noise to detect grinding burns, Sackmann et al. [25] performed a detailed analysis of the effective Barkhausen noise to improve the reliability of the Barkhausen noise multiparameter grinding burn analysis. In order to compensate for the shortcomings of Barkhausen noise detection and X-ray diffraction residual stress detection, Teixeira et al. [26] proposed a new method for the detection of grinding burns, i.e., thermal damage detection by Hall effect. Guba et al. [27] showed a method to detect the risk of gear grinding burns and thermal shock using surface layer modification charts without the need for experiments or simulations. This allows a rapid response to the adjustment of process parameters and is an efficient means of preventing grinding burns. Gao et al. [28] presents a novel grinding burn detection method basing on acoustic emission signals. Laser and grinding experiments were first performed to generate pure metal burn signals and grinding burn signals. Then, the cross wavelet transform and wavelet coherence were applied to reveal the coherence between the pure metal burn signal and the grinding burn signal. Grinding burns are always accompanied by changes in the microstructure, so the presence of grinding burns can be determined qualitatively by means of metallographic testing [29,30]. Changes in the microstructure also affect surface integrity parameters, the most important of which is the residual stress on the surface after grinding. Therefore, finding the relationship between residual stresses and grinding burns is also an important research task. Liu et al. [31] concluded that grinding burns can cause tensile stresses, which can cause serious damage to the surface fatigue resistance of the parts. and proposed a new method for grinding burn identification using a highly sensitive acoustic emission technique. Zhou et al. [32] carried out an experimental study on the surface integrity of rail grinding under humid conditions. It was shown that grinding burns and white etched layers, in particular, can be avoided in this environment and can bring the advantage of a 45% reduction in residual tensile stress. Xiao et al. [33] investigated the hazards associated with residual tensile stresses formed after grinding burns and proposed an analytical model based on the workpiece temperature profile for predicting the severity of residual stresses under various grinding cycles.

Although existing studies have made considerable achievements [34,35,36], further exploration is still needed due to the complex relationship between gear profile grinding burns and surface integrity. In addition, the subsurface layer properties alteration lacks detailed characterization and analysis. In this paper, we performed full-factor experiments on gear profile grinding and used thermocouples to measure the grinding temperatures at various locations on the tooth surfaces. The subsurface layer properties of ground tooth flank were characterized by Barkhausen noise measurement, X-ray diffraction, metallographic observation and Electron Back-Scattered Diffraction (EBSD). Finally, the subsurface layer properties alteration was described. In particular, the influence of the evolution of the microstructure on the residual stresses and the damage to the surface integrity caused by grinding burns.

## 2. Experimental Details

In this section, the details of gear profile grinding experiment were introduced. In addition, the tooth surface integrity was obtained including microstructure, residual stress, microhardness, and surface morphology.

### 2.1. Gear Profile Grinding Conditions

The gear profile grinding experiment was carried out on the KAPP NILES ZE400 gear grinding machine (KAPP NILES, Coburg, Bavaria, Germany), as shown in Figure 1. The gear parameters are listed in Table 1. The gear material is high-strength alloy steel (12 Cr2Ni4A), and its material composition is listed in Table 2. Table 3 shows the main thermophysical parameters of gear materials. In the experiment, the dynamic grinding temperature is measured using the embedded thermocouple method. The thermocouple is installed in blind holes close to the tooth flank to be processed, which are machined by electric spark drilling. The tooth flank to be processed and the tooth flank with blind holes are different sides of the same tooth. The distance between the bottom end of the blind hole and the gear tooth flank to be processed is about 150 μm. The thermocouples used in the experiment is a thermocouple of type 5TC-TT-K-40-36 produced by Omega Company (Norwalk, Connecticut, USA). In order to obtain a valid temperature value in the grinding area, the thermocouple needs to be placed close to the bottom of the blind hole. The thermocouple is then fixed with heat-resistant adhesive to prevent it from falling off during processing. Two thermocouples were installed at different positions; one thermocouple was close to the tooth top and the other was close to the tooth root to study the grinding temperature difference on the tooth flank. The data acquisition card with the model USB-1616HS was selected. Two channels were used to measure the temperature of the grinding zone in the experiment, and the sampling frequency was set to be 250 kHz. The cutting speed in the grinding process is extremely fast, and the grinding wheel quickly passes the thermocouple node, so the sampling frequency needs to be high enough to obtain the peak temperature value during the grinding process. As a result, a real-time grinding temperature can be obtained as the grinding wheel passes over the tooth surface area above the thermocouple.

To facilitate the drilling on the tooth flank, the teeth adjacent to the tooth flank to be drilled are cut off by wire-electrode cutting. The teeth on which the thermocouple is fixed and the corresponding machining tooth slots on the other side are reserved for experimental processing. The shape of the grinding wheel is adapted to the profile of the gear, but the condition of the wheel changes after each grinding process (e.g., shape, sharpness, etc.), so the wheel needs to be reshaped in the same way before each experiment to ensure the grinding wheel has the same initial condition.

The experiment condition was dry grinding, using a full factorial experimental scheme with controlled variables. The abrasive grain type of the wheel is ceramic grain (30% Cubitron 321). Wheel speed and normal feed depth are set as variables in the experiment, and other parameters remain unchanged. The specific experimental scheme is shown in Table 4.

### 2.2. Characterization of Subsurface Layer Properties

The surface integrity indicators studied in this paper include microstructure metallographic, residual stress, microhardness, and surface micromorphology. The processed gear teeth were cut off one by one by wire cutting to facilitate the above-mentioned surface integrity test, and an unprocessed gear was additionally selected as a blank control group. Figure 2 shows the detailed testing process and equipment. Cut out gears with a thickness of about 5 mm were used for mounting, and sandpaper from 300 to 2000 was used to polish the surface of the sample. Then, the samples were electropolished with a 5% perchloric acid alcohol solution. The electrolytic current was 0.8 A, the voltage was 30 V, and the electropolishing duration was 10 s. After 2 min of mechanical polishing, it was connected to clean water for half a minute, and the polishing liquid was silica suspension. After sample preparation, Electron Backscattered Diffraction (EBSD, EDAX, Pleasanton, CA, USA) testing and surface nanoindentation hardness testing were performed. Subsequently, the samples were corroded with a 4% nitric acid alcohol solution, and the surface metallographic phase was observed by electron microscopy. The remaining gear samples were ultrasonically cleaned and used for scanning electron microscopy (SEM, Thermo Fisher Scientific, Plainville, MA, USA) to capture the microscopic morphology of the processed tooth surfaces.

## 3. Results and Discussion

### 3.1. Determination of Grinding Burns

Pre-installed thermocouples are used to measure the grinding temperature at different locations on the tooth flank, which are located at *φ* = 0.42 rad and *φ* = 0.57 rad, respectively. As the thermocouple is located 150 µm below the gear surface, the grinding temperature picked up by the thermocouple will be slightly lower than the actual grinding temperature of the surface. The grinding temperature collected by the thermocouple is low-pass filtered to obtain the maximum measured grinding temperature for the corresponding grinding process parameters, as shown in Figure 3.

In gear profile grinding, the grinding contact area between the wheel and the tooth is a complex surface, so the normal grinding parameters in the grinding area are a function related to the involute. The parametric equation of the involute is established in the coordinate system of the gear profile grinding process as:(1)$$\{\begin{array}{c} x_{g}=r_{b}\left[\cos\left(-\beta \right)\left(\cos\left(\varphi \right)+\varphi \sin\left(\varphi \right)\right)+\sin\left(-\beta \right)\left(\sin\left(\varphi \right)+\varphi \cos\left(\varphi \right)\right)\right]\\ y_{g}=r_{b}\left[\cos\left(-\beta \right)\left(\sin\left(\varphi \right)+\varphi \cos\left(\varphi \right)\right)-\sin\left(-\beta \right)\left(\cos\left(\varphi \right)+\varphi \sin\left(\varphi \right)\right)\right] \end{array}$$
where *r_b_* is the base circle radius; *β* is half of the central angle corresponding to the base tooth pitch; *φ* = *θ_k_* + *α_k_*, *θ_k_* is the spread angle of involute; *α_k_* is the pressure angle of involute; *x_g_*-*y_g_* represents the coordinate system of gear profile grinding.

The distance between the wheel and the wheel axis at different locations in the tooth grinding area is the actual wheel radius, which is converted into the corresponding normal equivalent wheel radius on the tooth face according to the involute feature.
(2)$$R_{n}\left(\varphi \right)=\frac{R_{g\max}-\left[y_{g}\left(\varphi \right)-r_{b}\cdot \cos\beta \right]}{\cos\left(\arctan\left(\frac{dy_{g}\left(\varphi \right)}{dx_{g}\left(\varphi \right)}\right)\right)}$$
where *R_g_*_max_ is the maximum grinding wheel radius; *R_n_* is the normal equivalent wheel radius.

Likewise, the radial grinding depth is converted to the normal grinding depth relative to the tooth flank.
(3)$$a_{n}\left(\varphi \right)=a_{r}\cdot \cos\left(\arctan\left(\frac{dy_{g}\left(\varphi \right)}{dx_{g}\left(\varphi \right)}\right)\right)$$
where *a_n_* is the normal grinding depth.

Based on the above analysis of the tooth-flank–wheel-motion relationship, it can be seen that there is an uneven distribution of heat in the grinding area due to differences in the normal grinding parameters. As the rolling angle of the involute increases, the normal equivalent wheel radius decreases. The reduction in the normal equivalent wheel radius means that the wheel linear speed is also reduced. The normal grinding depth increases significantly with increasing the rolling angle of the involute. The combination of a high grinding depth and a low wheel speed, therefore, results in not only a high peak grinding temperature but also a significantly higher temperature gradient on the tooth surface. This results in a greater risk of grinding burns. Conversely, a smaller grinding depth matched with a higher wheel speed results in a lower grinding temperature, which helps to maintain the surface integrity of the gear. In terms of global parameters, the grinding depth is the most important factor influencing the peak grinding temperature, and increasing the grinding depth causes a significant increase in the grinding temperature. By increasing the wheel speed, the grinding temperature tends to increase and then decrease. This is because an increase in wheel speed generates more heat, but at the same time allows the wheel to carry more heat away. The Barkhausen noise test and the tooth flank residual stress test will further determine the range of parameters for grinding burns.

As shown in Figure 4, grinding burn tests and residual stress tests are performed on the completed gears under various processing parameters. The tooth flanks were scanned using a Rollscan 350 Barkhausen noise meter to obtain the corresponding MP values, and then the possible burned tooth flanks were determined by comparing the MP values of the tooth flanks under different process parameters. The X-ray diffraction method, which has the advantage of being non-destructive and having high testing accuracy, is employed to obtain residual stresses on tooth flanks. The Cr_K-Alpha X-ray tube was used for the measurement, and the average value was obtained after testing nine positions for each tooth flank.

The test results showed that the MP values measured by the Barkhausen noise meter had a more similar integrated process parameter correlation pattern with the tooth surface residual stress values in the full factor experiment of grinding-depth–wheel-speed. The lower MP values and larger residual compressive stresses are distributed in the parameter range of shallow grinding depth and high wheel speed. On the contrary, the larger MP values and the smaller residual compressive stresses are clearly concentrated in the parameter range of large grinding depth and low wheel speed. The MP value is a dimensionless parameter, and the tooth flank where grinding burns may occur is determined by comparing all the MP values obtained. Normally, the MP value of the tooth flank with reduced hardness after grinding will be too large and, thus, tempering burn is considered to have occurred. At the same time, temper burn will destroy the initial residual compressive stress field of the tooth flank and replace it with a smaller residual compressive stress. In summary, grinding depths of more than 20 μm matched with wheel speeds below 30 m/s will result in severe grinding burns. Although the MP value has a strong correlation with the residual stress, it cannot essentially explain the mechanism of the evolution of the residual stress [37]. Therefore, to essentially reveal the influence of grinding behavior on the alteration of the residual stress state of tooth flanks requires a deeper insight into the evolution of the microstructure.

### 3.2. Microstructure Evolution and Burn Mechanism

Figure 5 shows the EBSD results of the microstructure of the studied high-strength aerospace gear before grinding, i.e., the initial state. In contrast to the core, the surface layer of the gear is carburized and quenched to obtain a fine martensitic structure with high hardness and wear resistance. At the same time, the gear core retains good impact toughness. It was measured that the tooth flank had a compressive residual stress of about −350 MPa before grinding. This microstructure allows the gear to easily cope with high speed and heavy load operating environments and to have ideal fatigue resistance. However, the thermal behavior of the grinding process can affect the microstructure of the gear surface, causing its performance to deviate from the design requirements and even causing the tooth surface to fail. Therefore, it is necessary to minimize the damage caused to the tooth flank by the grinding process.

The microstructure of the tooth flank after grinding with different process parameter configurations shown in Figure 6 has obvious differences, thus indicating that the tooth flank undergoes different phase transformation processes under the two working conditions. Figure 6a showed obvious signs of grinding burns, with different depths of burns at different locations on the tooth surface, which were related to the random distribution of abrasive grains and differences in the normal grinding parameters of the tooth flank, and the maximum depth of burns could be more than 100 μm deep into the tooth surface. The surface of the gear after acid corrosion did not show a bright white layer but patches of black areas, indicating that the grinding temperature did not reach the critical value of the second quenching burn, so the tooth flank experienced tempering grinding burn. The tissue in the area where the grinding burns occurred is mainly unevenly distributed tempered sorbite, with a large amount of martensite still present in the unburned area. The surface in Figure 6c shows no signs of burn, and the surface tissue is predominantly martensitic, but there is a small amount of tempered martensite in some areas near the tooth flanks. The EBSD results of the gears within 100 μm of the surface layer after the grinding process (Figure 6b,d) were compared with the microstructure of the initial state (Figure 5a). The microstructure and grain size of grinding burns were significantly changed, and it was almost difficult to see the obvious martensite tissue and, instead, there were many small equiaxed grains. In contrast, the surface layer of the tooth without grinding burns was very close to the initial state, and the martensitic organization was clearly visible in the form of slats at 60° or 120° to each other. It can be seen that the harm caused by grinding burns is the destruction of the otherwise favorable microstructural state of the gear surface.

### 3.3. Subsurface Layer Properties Alteration in Burn Process

Grain statistics based on EBSD results and selected representative grain distribution states are presented in Figure 7. The average equivalent circle diameter of the grains before grinding was 1.3 μm, and that of the unburned tooth flank was 1.2 μm, which were relatively similar overall, but the grains smaller than 2 μm would increase slightly after the grinding process. In contrast, the surface grains of the gears with tempered grinding burns were significantly smaller overall, with an average equivalent circle diameter of only 1 μm. More notably, after the occurrence of tempered grinding burns, the percentage of grains with an equivalent circle diameter greater than 5 μm decreases substantially. The triangular-shaped boundary of the original martensitic composition can be judged from the grain orientation as shown by the white dashed line in Figure 7. Compared with the martensite tissue before grinding, the tissue morphology without grinding burns did not change much, and the slate-like martensite grains remained the same. However, the original martensite organization was replaced by many fine equiaxed grains inside the tempered grinding burns, and the slate-like grains were hard to find.

To further elucidate the microstructure evolution caused by tempered grinding burns, the intergranular boundary angles were calculated from the grain orientation difference. The grain boundaries larger than 15° were defined as large-angle grain boundaries, and those between 2° and 15° were defined as small-angle grain boundaries, as shown in Figure 8. The results showed that the percentage of grain boundaries with large angles after tempered grinding burns reached 76.6%, compared with 69.9% for unburned ones, which shows that tempered grinding burns caused a certain degree of tissue recovery and recrystallization. During the grinding process, the metal will recover and recrystallize due to the force and thermal load. The recovery is a process in which the dislocations inside the deformed grains are polygonalized and further transformed into equiaxed subcrystals, and subgrain boundaries gradually appear inside the original deformed grains during the polygonalization process, and the difference in grain boundary orientation between adjacent subcrystals is generally between 2–15°. The driving force of recrystallization is the unreleased deformation storage energy after recovery, and the nucleation mechanisms include the grain boundary bow-out mechanism, subcrystal migration mechanism, and subcrystal merger mechanism. After recrystallization, the orientation difference between adjacent grains increases further and becomes a large-angle grain boundary (greater than 15°) [38].

The microstructure evolution of the gears in this study during tempered grinding burn is as follows: the carbon in the initial martensite tissue starts to precipitate at high grinding temperatures; the ferrite, which is formed by the decomposition of martensite, starts to maintain the martensitic slat morphology, but under the action of high grinding temperatures and grinding forces, the ferrite transforms to a more stable equiaxed fine ferrite morphology; thus, the martensitic ferrite recovers and recrystallizes. This evolution of the microstructure also leads to a change in the corresponding residual stress state. The initial compressive stress on the tooth flank has been greatly relieved after the recovery and recrystallization of the tempered grinding burned tissue, resulting in the final processed tooth flank with almost no compressive stress. The unburned tooth flank retains a large amount of the pre-grinding tissue morphology and, therefore, retains a considerable amount of compressive residual stress.

Based on the analysis of the evolution of the grinding burn mechanism, it is known that the recrystallization caused by the tempering burn on the tooth surface will also decrease the hardness of the corresponding burned layer. Due to the shallow depth range, the hardness of the burned layer was measured using a nanoindenter and, as a control, the hardness of the unburned gear surface layer was also measured. The experimentally measured nano-hardness of the tooth is shown in Figure 9. The hardness of the surface layer of the gear without grinding burns tends to decrease gradually along the tooth flank, with a maximum nano-hardness of 12.01 GPa on the tooth flank. Compared to the unburned tooth, the nano-hardness after experiencing tempered grinding burns decreased substantially, with a tooth flank nano-hardness of only 7.75 GPa. The closer to the tooth flank, the greater the degree of nano-hardness decline, resulting in the lowest hardness on the tooth flank and a small increase in hardness distribution down the tooth flank. The average nano-hardness of normal tooth was 10.49 Gpa, and the average nano-hardness of tooth after the occurrence of tempered grinding burns was 8.11 Gpa, which was 22.7% lower than that of unburned. The experimental results are consistent with expectations and provide further evidence for the plausibility of the residual stress evolution mechanism of gear profile grinding illustrated in this paper.

The damage caused by grinding burns to the surface integrity of gears is not only in terms of residual stress and hardness, but also to the microscopic morphology of the tooth flank, as shown in Figure 10. The transient high temperature generated by the contact between the grains and the tooth flank is mainly in the plowing stage, where the more prominent grinding grains rub against the workpiece, more heat will be generated. The oxide formed by the transient high temperature of the grinding point causing burns is very easy to peel off the tooth flank, thus, forming tiny defects near the grinding groove with a large depth on the tooth flank. The results also showed that the area and density of such micro-defects were significantly larger near the top of the tooth, where the grinding temperature was higher, than at the root of the tooth, where the grinding temperature was lower. The difference at the top and bottom of the gear teeth can lead to uneven surface integrity in the actual gears produced, which further affects the load carrying capacity and surface fatigue resistance of the gears. These damages are often more severe at the top of the teeth. Therefore, in addition to efforts to reduce the grinding temperature, further process optimization of the top area should be considered, for example, by leaving less machining allowance in the top area of the teeth.

## 4. Conclusions

In this paper, we conducted a comprehensive study to understand the grinding burn behavior in gear profile grinding with complex process geometry. The relationship of subsurface layer properties, grinding burn and grinding temperature of gear tooth is established. Note that the type of the grinding burn is tempering burn according to the experimental conditions and gear grinding parameters. The main conclusions obtained are as follows:(1)The increase in radial grinding depth is the most important factor leading to the increase in peak grinding temperature. With the increase in wheel speed, the peak grinding temperature showed a trend of a small increase and then a decrease. The temperature gradient between different rolling angle of the involute positions on the tooth flank was greatly influenced by the combination of grinding parameters. Radial grinding depths of more than 20 μm matched with wheel speeds below 30 m/s will result in severe grinding burns. The risk of burn can be significantly reduced at greater wheel speeds and lower radial grinding depth.(2)The EBSD results of grinding burn tooth show that the proportion of the large angle grain boundary is increased by 6.7% compared to the initial tooth. It suggests that the recrystallization occurs when the tempering burn happens. Tempering burn also results in the breakdown of high strength martensite and the formation of inhomogeneous secondary tempered sorbite.(3)The relationship between microstructure, grinding burn, and the mechanical properties of gear tooth is established. The substantial reduction in compressive residual stress and nano-hardness could be explained by the recovery and recrystallization of the tooth surface microstructure. Severe grinding burns can form tiny defects in the tooth surface near the tooth grinding grooves during the plowing stage.

## Figures and Tables

**Figure 1 materials-16-02493-f001:**
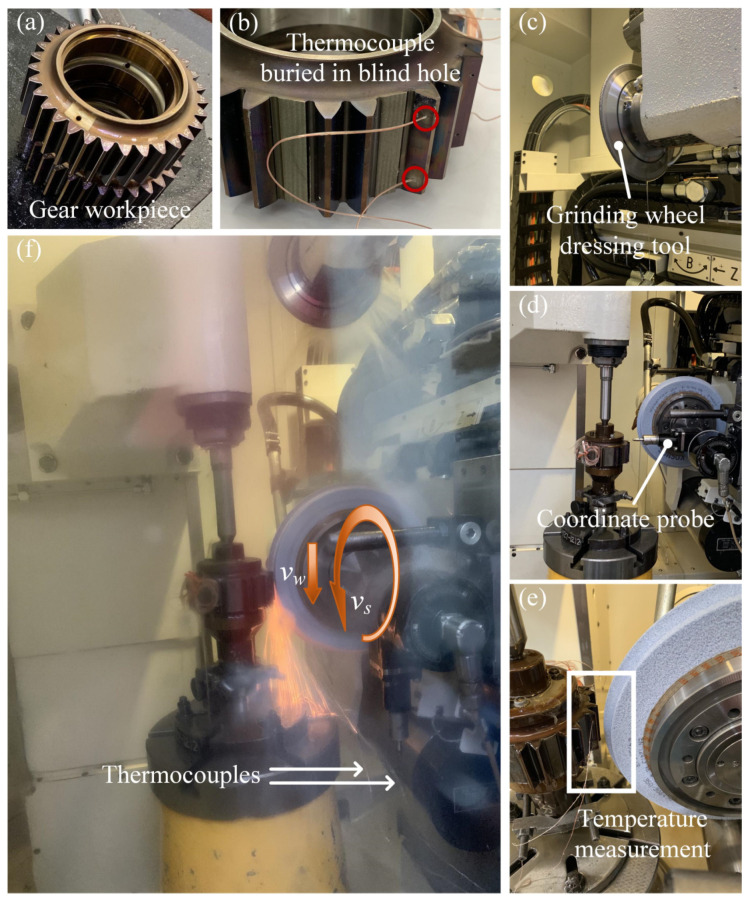
Gear forming grinding equipment: (**a**) experimental gear; (**b**) installed thermocouple; (**c**) wheel dressing; (**d**) coordinate probe; (**e**) grinding temperature measurement; (**f**) machining process.

**Figure 2 materials-16-02493-f002:**
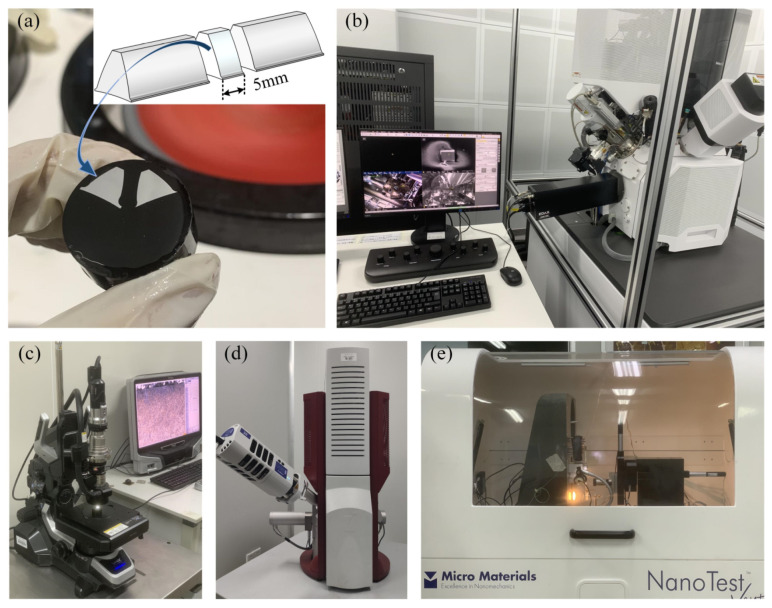
Testing experiment of gear surface integrity after machining: (**a**) metallographic sample preparation. (**b**) Electron Backscattered Diffraction (EBSD) measurement of the microstructure. (**c**) Super-depth-of-field microscopy to observe the surface metallography. (**d**) Scanning electron microscope (SEM) test of the processed surface. (**e**) Nanoindentation hardness test of tooth flank.

**Figure 3 materials-16-02493-f003:**
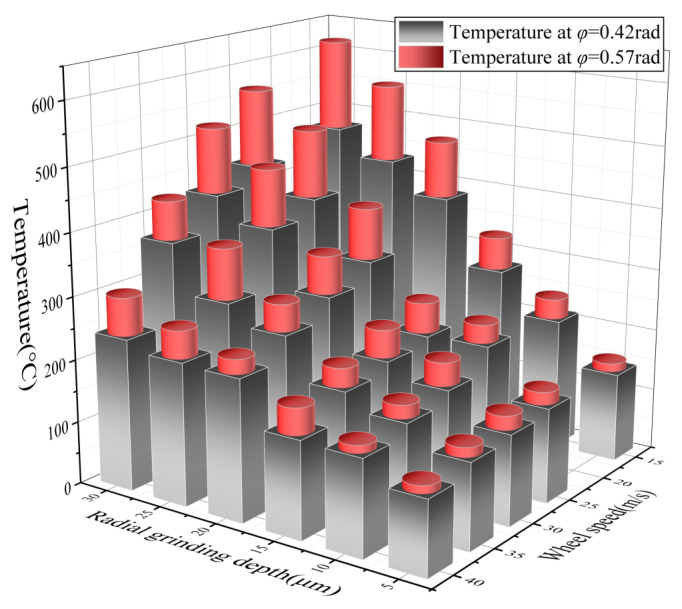
Grinding temperature of tooth.

**Figure 4 materials-16-02493-f004:**
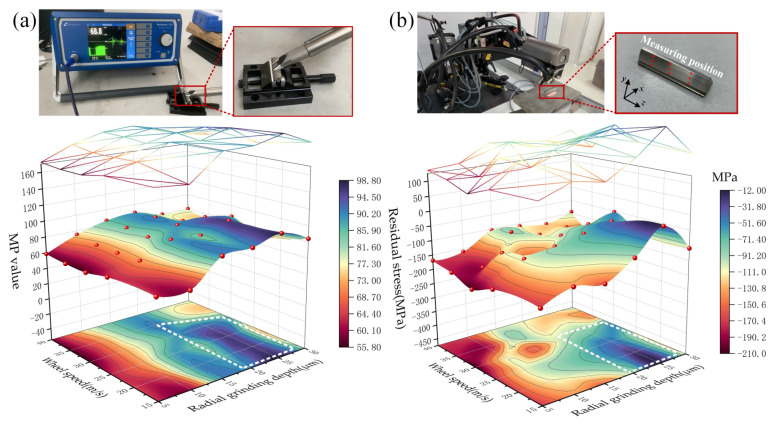
Testing and results: (**a**) Barkhausen noise test and results; (**b**) tooth flank residual stress test and results.

**Figure 5 materials-16-02493-f005:**
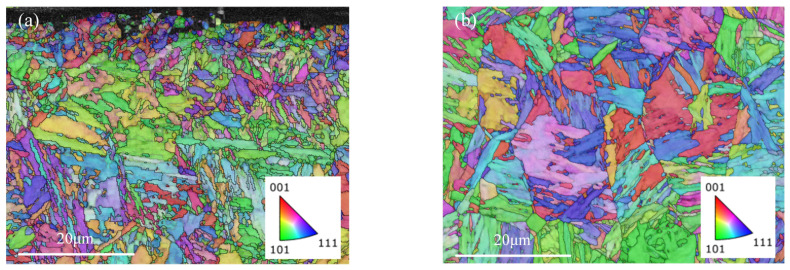
EBSD results of the microstructure of gears before grinding: (**a**) Surface of the gear; (**b**) core of the gear.

**Figure 6 materials-16-02493-f006:**
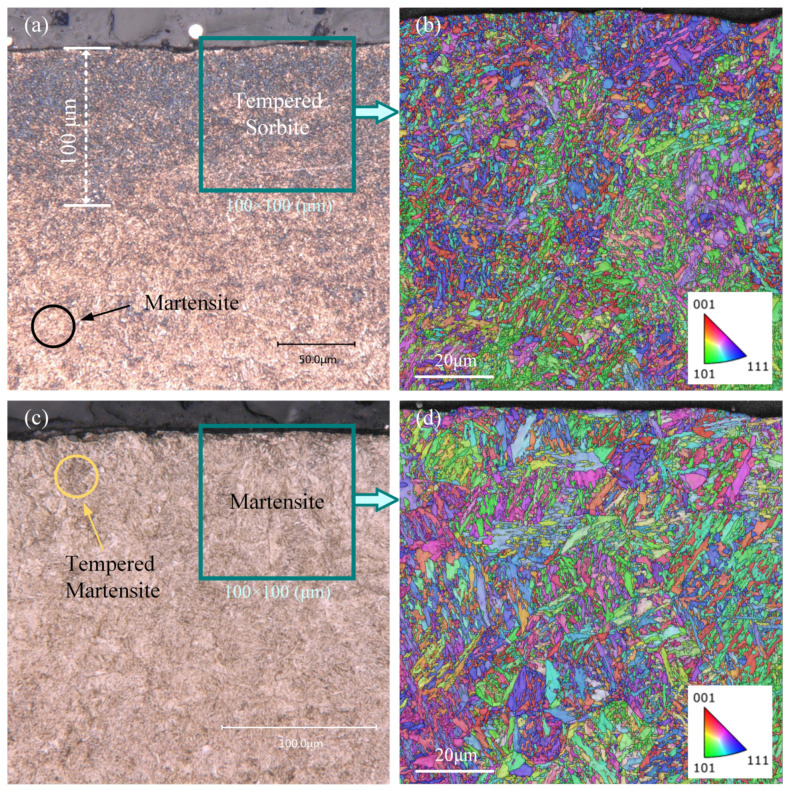
Metallographic and microstructure of the gear after grinding: (**a**) surface metallography of gears at a wheel speed of 15 m/s and a grinding depth of 25 μm; (**b**) EBSD results of the surface microstructure of gears at a wheel speed of 15 m/s and a grinding depth of 25 μm; (**c**) surface metallography of gears at a wheel speed of 30 m/s and a grinding depth of 5 μm; (**d**) EBSD results of the surface microstructure of gears at a wheel speed of 30 m/s and a grinding depth of 5 μm.

**Figure 7 materials-16-02493-f007:**
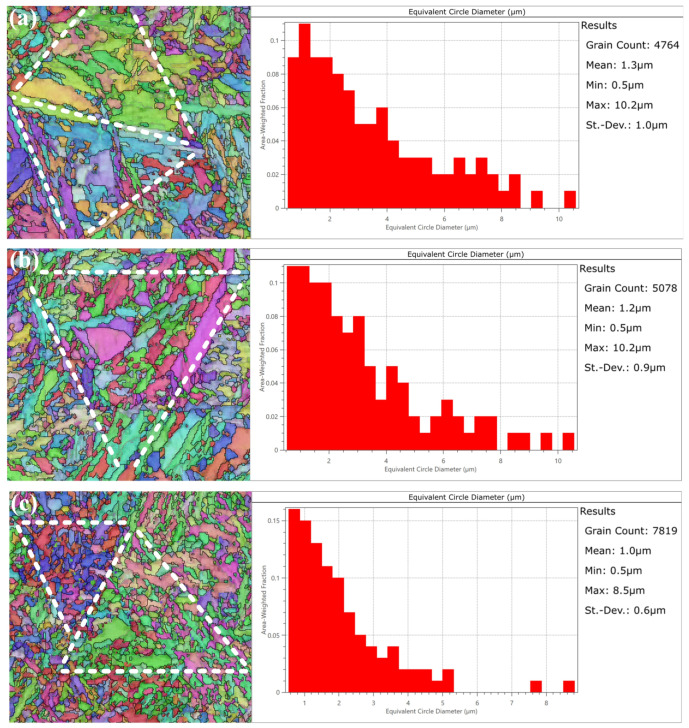
Grain size statistics and distribution patterns: (**a**) initial state; (**b**) surface layer of the tooth without grinding burns; (**c**) surface layer of the tooth with tempered grinding burns.

**Figure 8 materials-16-02493-f008:**
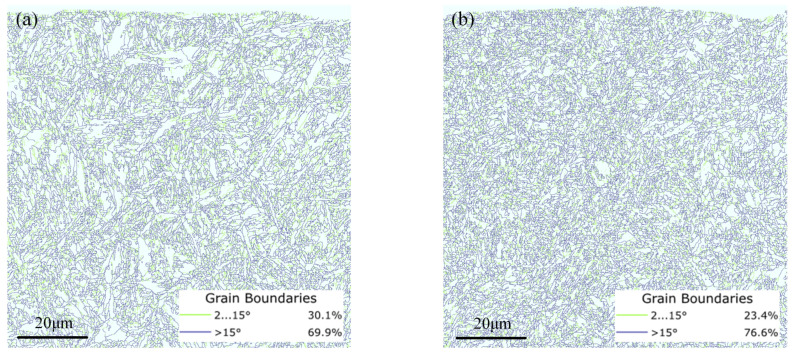
Grain boundary angle calculations: (**a**) surface layer of normal gear; (**b**) surface layer of tempered grinding burned gear.

**Figure 9 materials-16-02493-f009:**
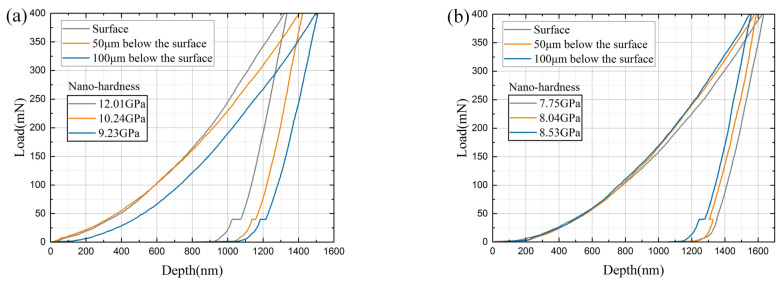
Nano-hardness of the gear surface layer: (**a**) nano-hardness of normal tooth; (**b**) nano-hardness of tooth after tempered grinding burns.

**Figure 10 materials-16-02493-f010:**
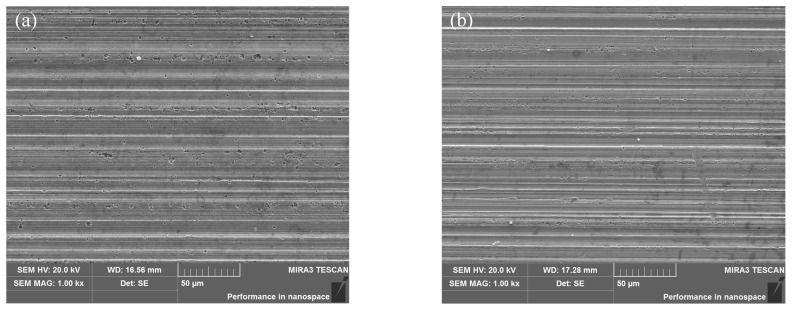
SEM of tooth flank after tempered grinding burn: (**a**) top of the tooth; (**b**) root of the tooth.

**Table 1 materials-16-02493-t001:** Gear parameters.

Gear Parameter	Value
Modulus (mm)	3.8788
Number of teeth	31
Pressure angle (°)	28
Flank width (mm)	45

**Table 2 materials-16-02493-t002:** Composition of 12Cr2Ni4A alloy steel.

Element	C	Mn	Si	S	P	Cr	Ni
Composition (%)	0.1~0.15	0.3~0.6	0.17~0.37	≤0.015	≤0.025	1.25~1.75	3.25~3.75

**Table 3 materials-16-02493-t003:** Thermophysical parameters of the gear.

Material	*k* (W/(m·K))	*c* (J/(kg·K))	*ρ* (kg/m^3^)	*α* (1/K)
12Cr2Ni4A	28	547	7840	1.3 × 10^−5^

**Table 4 materials-16-02493-t004:** Experimental parameters.

Processing Parameter	Value
Feed speed (mm/min)	1200
Maximum diameter of wheel (mm)	286
Abrasive type	Ceramic grain (30% Cubitron 321)
Abrasive size	80
Radial grinding depth (μm)	5 10 15 20 25 30
Wheel speed (m/s)	15 25 30 35 40

## Data Availability

All data that support the findings of this study are included within the article.

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
