# Peer review of "Towards Understanding Subsurface Characteristics in Burn Process of Gear Profile Grinding"

_materials, 2023, doi:10.3390/ma16062493_

Round 1

Reviewer 1 Report

The paper discusses the very interesting problem of finishing the side of the gear teeth by grinding. During grinding, burning of the surface can occur, which affects the exploitation of the gears. The authors need to make some corrections and provide explanations according to the comments below.

 1. It is necessary to explain in more detail how to measure temperatures with thermocouples (picture 1b and picture 1e), when the tappet is engaged with the side surface of the gear teeth. Why wasn't the temperature measured at some point between the top and bottom of the side of the gear tooth?

 2. On the basis of what criteria is the range of processing speeds (15 - 40 m/s) and machining depth (5 - 30 µm) defined? Will the conclusion "The risk of burns can be significantly reduced at higher spindle speeds and smaller cutting depths" also apply when the feed speed is different from 1200 mm/min?

 3. The research showed that there is a difference, as a result of burns, structure, and residual stresses, at the top and bottom of the gear teeth. What effect does it have on the behavior of the gears in exploitation?

 4. In Figure 10, 10b is listed first, and then 10a. Is that a mistake?

5. Within the list of references, [J] appears in all references. What is it supposed to represent?

Reviewer 2 Report

The presented work from the scientific approach is very interesting and up-to-date. The analysis of the gear grinding process is very important from the point of view of obtaining the appropriate operating parameters of the gear, the authors pay attention to this and carry out a comprehensive analysis of the surface structure study.
The work omitted the aspect of machining accuracy, from the point of view of the requirements for grinded gears (class 3-4) there is a sense of insufficiency. However, with regard to the subject of the analysis and, as a consequence, the research carried out, the work in the assumed scope is complete.The analysis of the literature should be supplemented with issues related to the design of the transmission, work, defects and wear, as this affects the phenomenon undertaken in the study. I encourage you to familiarize yourself with the works: Diagnosis of the Operational Gear Wheel Wear, Analysis of Modification of Spur Gear Profile, I also recommend the authors to write an introductory word introducing the reader to the gear technology of their meshing, in this context it is important to justify the topic taken because from the utilitarian point of view it is important and has a strong relationship with the technology of gears - gears as themselves.

Some details in the work need to be explained in more detail in the text:
Line 22 Abstract: term: "The risk of burn can be significantly reduced at greater wheel speeds and lower cutting depths." What does the term "cutting depths" refer to? Did the author mean machining allowances? This needs to be standardized throughout the text Line 143, grinding wheel has been dressed to "ensure the constant state" what does this mean? the involute equation is presented, of course it corresponds to the outline of the toothed wheel, however, it requires an explanation of how the grinding wheel used in the process was profiled?
It is very important in grinding in particular complex profiles to use a compatible tool, the division of allowances as an equidistant must be constant over the entire height of the profile (tooth profile).
Please, explain the exact kinematics of grinding and division of allowances, ............. line 204 "As the involute parameter increases,....." linking the circumferential speed of the grinding wheel with the involute parameter is quite a big simplification .....it is obvious that it is related to the diameter of the tool.
Also... line 207 "The normal grinding depth increases significantly with increasing involute parameters." What parameters of the involute did the author have in mind?
What is meant by "grinding depth" maybe it's moving the tool towards the root area? This needs some clarification.
Figure, 9 - nano-hardness, is the term "Depth" correct? should it be "displacement"? maybe more "depth deviation"?
consequently line 388, "was proportional to the involute parameter"...... as well grinding depths....... from a technological point of view, this is a big generalization.
